# Polar Codes and a M-ary Modulation-Based OFDM-PLC System

**Ziyi Wang, Yichen Wang and Rui Chen \***

Department of Communication Engineering, Nanjing Institute of Technology, Nanjing 211167, China; Y00450210506@njit.edu.cn (Z.W.); Y00450220519@njit.edu.cn (Y.W.)
\* Correspondence: chenrui@njit.edu.cn

**Abstract:** Power Line Communication (PLC) serves as a medium for communication over power lines, utilizing the existing power grid for information transmission. It offers a low-cost, highly scalable signal transmission method and has the potential to become the preferred technology for providing broadband in smart homes, offices, and smart grids. However, noise in the power line channel, especially impulse noise, seriously affects the communication quality of PLC. In this paper, we establish an OFDM-PLC system with higher-order modulation and polar codes. The higher-order QAM or APSK modulation technique is employed to increase the signal transmission rate and the system performance is analyzed. To combat the impulse noise in the PLC channel, we first model it using a superposition of multi-damping sinusoidal functions and then introduce the polar coding scheme to suppress the impulse noise in the system, thereby improving the transmission reliability. Simulation results verify that the proposed polar coding scheme based on the OFDM-PLC system can improve the Bit Error Rate (BER) performance of PLC channel transmission under impulse interference.

**Keywords:** PLC; polar code; impulse noise; higher-order QAM; higher-order APSK

## 1. Introduction

Power Line Communication (PLC) uses existing distributed power transmission networks for data transmission and information exchange. As a communication access technology, it can easily connect devices to the internet without the need for dedicated lines. Compared to other wired communication technologies, PLC has advantages such as wide coverage, convenient networking, strong scalability, low cost, and "plug-and-play" capabilities, making it suitable for applications in remote meter reading, mine communication, smart homes, and power IoT [1–3]. However, since power lines are not designed as communication channels, PLC faces several major challenges in practical applications. These challenges arise from the adverse conditions of power lines, such as frequency-selective fading [4] and impulse noise [5,6], which can distort the received signal. In [7], the authors pointed out that noise is the most critical element affecting the PLC communication process, and they divided noise into background noise and impulse noise. In order to reduce the impact of noise on PLC, it is necessary to establish an appropriate noise model to accurately characterize the noise. Zimmermann M et al. [8] analyzed the spectrum and time-domain characteristics of impulse noise in typical power line environments, studied the impulse rate and interference rate in these environments, and proposed a segmented Markov chain random impulse noise statistical model. Guo et al. [9] modeled the background noise in the power line channel using singular value decomposition and LD recursion to calculate its AR model parameters. However, they only studied one type of noise in the power line channel, and power line noise has non-stationary, non-Gaussian, and characteristics of different types of noise superimposed, which cannot reflect the overall noise characteristics. A cyclic smooth model was proposed in [10], which adjusted the instantaneous power spectral density of channel noise to adapt to experimental measurement data. This model has fewer coefficients and can simulate ideal power line noise in the case of higher orders.

To characterize the characteristics of various types of noise, the Middleton-A model is the most popular non-Gaussian noise model, which is composed of narrowband impulse noise and Gaussian noise. Many researchers use this type of noise model when studying PLC channels [11]. However, Middleton-A class noise, as a human-made impulse construction method, is composed of independent sampling points, which cannot reproduce the physical characteristics of impulse noise oscillation attenuation, and therefore needs further improvement.

On the other hand, with the continuous development of modern digital signal processing technologies such as modulation and channel coding, Orthogonal Frequency Division Multiplexing (OFDM) technology is considered to be an effective method for solving problems such as noise, multipath propagation, and frequency-selective fading in PLC [12]. In [13], the performance of using OFDM in PLC channels was analyzed and compared with single-carrier modulation systems, and the authors pointed out that OFDM outperformed single-carrier modulation in terms of BER performance. For channel coding, the authors in [14] proved that modeled PLC channels can have good image transmission capabilities with the help of LDPC coding. In [15], the authors compared the performance of LDPC codes and Turbo codes in the PLC environment and proved that the performance of LDPC codes can approach that of Turbo codes with higher block lengths. In [16], OFDM technology with subcarrier mapping using BPSK and QAM achieved excellent performance by introducing LDPC codes in error correction coding. Recently, after Arikan et al. invented polar codes [17], many studies have shown that polar codes can achieve good performance when applied to PLC systems with impulse noise. Niu K et al. [18] evaluated the BER performance of polar codes against impulse noise, but only in single-carrier scenarios. Furthermore, they solved the problem of limited single-carrier scenarios in [18] by introducing polar coding in OFDM-based multi-carrier PLC systems and discussing Successive Cancellation (SC) decoding techniques [19]. Based on this, two decoding methods, Successive Cancellation List (SCL) and Successive Cancellation Stack (SCS), were proposed to improve the performance of polar codes. In addition, A. Hadi et al. [20] pointed out that soft decision decoding is superior to hard decision decoding, and performance can be further improved with increasing code word length. They also proved that polar codes have stronger capabilities than other codes in correcting errors caused by impulse noise in PLC systems.

In this paper, we propose an OFDM-PLC system with higher-order modulation and polar encoding modules under impulse noise. In order to describe the impulse noise more accurately, a multi-damping sinusoidal function is used to model the impulse noise in the PLC channel. A polar code encoding scheme with low complexity is used to suppress the impulse noise and improve transmission performance. Furthermore, high-order 16QAM or 16APSK modulations and soft demodulation techniques are introduced in the OFDM-PLC system, which shows improved performance and are suitable for PLC channels, thereby enhancing transmission efficiency. The remainder of this paper is organized as follows. Section 2 introduces the encoding and decoding principles of polar codes. Section 3 describes the OFDM-PLC system model based on polar codes, including the structural diagram, multipath channel model, and noise model. Section 4 introduces the modulation and demodulation principles of higher-order QAM and APSK, and applies them to the OFDM-PLC system. Section 5 presents the simulation and performance evaluation of the system. Finally, conclusions are drawn in Section 6.

## 2. Polar Codes

Polar codes are obtained through channel polarization processing, which aggregates and separates bits on the coding side to present different reliabilities of each sub-channel. As the code length approaches infinity, some channels tend to be noise-free channels with a channel capacity of one for transmitting information bits, while the other channels tend to be all-noise channels with a channel capacity of zero for transmitting frozen bits and auxiliary information bits. On the decoding side, the polarized channels can be decoded

with suitable decoding methods at lower complexity to achieve performance close to that of maximum likelihood decoding.

*2.1. Encoding*

Polar codes, as a type of linear block codes, can be represented by encoding parameters $(N, K, A, u_{A^c})$. The encoder uses the channel polarization phenomenon to perform channel coding, making it possible to achieve symmetric channel capacity. Through estimating the reliability of polarized channels and performing Gaussian approximation to sort the sub-channels by confidence, reliable sub-channels with high reliability are selected to map $K$ information bits $u_A$, while low-reliability sub-channels are mapped frozen bits $u_{A^c}$, which are often taken as an all-zero sequence. $u_A$ and $u_{A^c}$ form a complete sequence $u_1^N : u_1, u_2, \ldots, u_N$, with a code length of $N = 2^n$. Multiplying this sequence by the generator matrix $G_N$ of the polar code yields the encoded bit sequence $x_1^N : x_1, x_2, \ldots, x_N$, which is represented as:

$$x_1^N = u_1^N G_N \tag{1}$$

where the generator matrix $G_N$ is:

$$G_N = B_N F^{\otimes n}, \; n = \log_2 N \tag{2}$$

where $B_N$ is the bit-flip permutation matrix, $B_N = R_N(I_2 \otimes B_{N/2})$, $R_N$ is the parity interleaved, and $F^{\otimes n}$ denotes the Kronecker product $n$ times, and $F = \begin{bmatrix} 1 & 0 \\ 1 & 1 \end{bmatrix}$.

*2.2. Decoding*

At the decoder, the SC decoding algorithm first partitions the decision units. If a decision bit is a frozen bit, its value is determined based on the known frozen bit value. Otherwise, its value is determined through the decision function $h_i$, and the decision rule is:

$$\hat{u}_i \triangleq \begin{cases} u_i, \; i \in A^c \\ h_i\left(y_1^N, \hat{u}_1^{i-1}\right), \; i \in A \end{cases} \tag{3}$$

The decision function $h_i$ is defined as:

$$h_i\left(y_1^N, \hat{u}_1^{i-1}\right) \triangleq \begin{cases} 0, \; L_N^i\left(y_1^N, \hat{u}_1^{i-1}\right) \geq 0 \\ 1, \; otherwise \end{cases} \tag{4}$$

where $L_N^i\left(y_1^N, \hat{u}_1^{i-1}\right)$ is the log-likelihood ratio (LLR) for each channel, which is calculated by:

$$L_N^i(y_1^N, \hat{u}_1^{i-1}) \triangleq \ln \frac{W_N^i\left(y_1^N, \hat{u}_1^{i-1}\big|0\right)}{W_N^i\left(y_1^N, \hat{u}_1^{i-1}\big|1\right)} \tag{5}$$

where $W_N^i\left(y_1^N, \hat{u}_1^{i-1}\big|u_i = 0/1\right)$ represents the transition probability of the $i$ separated sub-channel.

The LLR $L_N^i\left(y_1^N, \hat{u}_1^{i-1}\right)$ is recursively calculated by:

$$\begin{cases} L_N^{2i-1}\left(y_1^N, \hat{u}_1^{2i-2}\right) = f\left(L_{N/2}^i\left(y_1^{N/2}, \hat{u}_{1,o}^{2i-3} \oplus \hat{u}_{1,e}^{2i-2}\right), L_{N/2}^i\left(y_{N/2+1}^N, \hat{u}_{1,e}^{2i-2}\right)\right) \\ L_N^{2i}\left(y_1^N, \hat{u}_1^{2i-1}\right) = g\left(L_{N/2}^i\left(y_1^{N/2}, \hat{u}_{1,o}^{2i-2} \oplus \hat{u}_{1,e}^{2i-2}\right), L_{N/2}^i\left(y_{N/2+1}^N, \hat{u}_{1,e}^{2i-2}\right), \hat{u}_{2i-1}\right) \end{cases} \tag{6}$$

where $\hat{u}_{1,o}^{2i-3}$ represents all odd numbers between 1 and $2i-3$, $\hat{u}_{1,e}^{2i-2}$ represents all even numbers between 1 and $2i-2$, $i \in [1, N]$, functions $f$ and $g$ are defined as follows:

$$\begin{cases} f(a,b) \triangleq \ln\left(\frac{e^{a+b}+1}{e^a+e^b}\right) \approx sgn(ab)\min(|a|,|b|) \\ g(a,b,c) \triangleq b + (-1)^c a \end{cases} \tag{7}$$

As the SC decoding algorithm uses LLR decision-making based on bit-by-bit, it can lead to error propagation due to incorrect decisions at certain nodes, which cannot guarantee maximum likelihood decoding of the entire codeword. To further improve decoding performance, the SCL algorithm [21] calculates path metrics (PM) values for $2N$ paths (with a code length of $N$) when performing LLR decision-making on information bits. When the path width is $L$ and $L \geq 2$, path selection is performed based on the PM values, and the optimal path is selected as the decoding result. The PM is calculated as follows:

$$PM_l^i = \begin{cases} PM_l^{i-1}, \ if \ u_i \in A \ or \ u_i \in A^c \ and \ \hat{u}_j[l] = \delta\left(L_n^j[l]\right) \\ PM_l^{i-1} + \left|L_N^j[l]\right|, \ if \ u_i \in A \ or \ u_i \in A^c \ and \ \hat{u}_j[l] \neq \delta\left(L_n^j[l]\right) \\ +\infty, \ if \ u_i \in A^c \ and \ incorrect \ value \end{cases} \tag{8}$$

where $i \in \{1, 2, \dots, N\}$ and $l$ represent the corresponding paths, $l \in \{1, 2, \dots, L\}$ and $PM_l^0 = 0$; the LLR of $L_N^j[l]$ is $L_N^j[l] = \ln\frac{W_N^i(y_1^N, \hat{u}_1^{i-1}[l]|0)}{W_N^i(y_1^N, \hat{u}_1^{i-1}[l]|1)}$; $\delta(x) = \frac{1}{2}(1 - sgn(x))$ represents the decision bit $\hat{u}_i$ as $\hat{u}_i = \delta\left(L_N^i\left(y_1^N, \hat{u}_1^{i-1}\right)\right)$.

To improve the accuracy of path selection in the SCL algorithm, the CA-SCL algorithm introduces cyclic redundancy check (CRC) codes [22], forming a CRC-polar concatenated code. The decoding algorithm flowchart is shown in Figure 1 below:

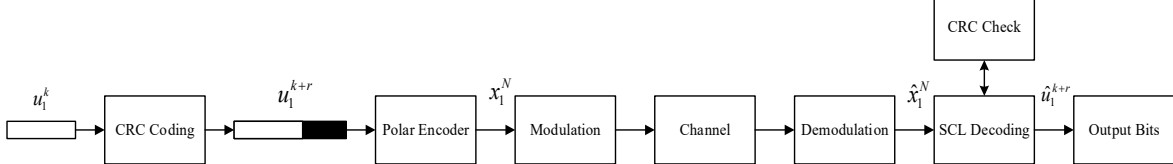

**Figure 1.** CA-SCL Decoding Algorithm.

The original bit stream $(u_1, \dots, u_k)$ and the CRC check bits with the length of $r$ form sequence$(u_1, \dots, u_k, u_1, \dots, u_r)$, which is polar encoded to obtain the codeword sequence $(x_1, \dots, x_N)$. The information bits $(x_1, \dots, x_N)$ are modulated and demodulated and input into the SCL decoder as the sequence $(\hat{x}_1, \dots, \hat{x}_N)$ (the modulation and demodulation module is described in 2.1), and the SCL decoder provides $L$ estimated information bits $(\hat{u}_1, \dots, \hat{u}_k, \hat{u}_1, \dots, \hat{u}_r)_l$, $i \in \{1, 2, \dots, L\}$, corresponding path metrics $PM_l$, and $i \in \{1, 2, \dots, L\}$. If none of the $(\hat{u}_1, \dots, \hat{u}_k, \hat{u}_1, \dots, \hat{u}_r)_l$, $i \in \{1, 2, \dots, L\}$ passes the CRC check, the decoding fails. If only one of them passes the CRC check, the estimation that passes the CRC check is chosen as the decoding result. If multiple estimations pass the CRC check, the estimation with the minimum metric value is selected as the decoding result among those that pass the CRC check.

## 3. System Model

### 3.1. Structural Diagram

This paper proposes an OFDM-based PLC polar coding system, and the structure diagram is shown in Figure 2. A random binary bit sequence $u$ is input into the polar encoding module, and the encoded sequence $x$ is interleaved with bits and passed to the modulation mapper, where a high-order modulation scheme that combines phase

and amplitude, such as 16PSK, 16QAM, and 16APSK, is used to improve bandwidth transmission efficiency. The subcarrier modulated signal sequence $S$ is generated according to the modulation scheme for OFDM system modulation and then converted into $N$ parallel subcarrier modulation signals with a speed of $1/T_s$ ($T_s$ is the signal duration) through serial-to-parallel conversion. These signals are represented as $S_0 \sim S_{N-1}$. The parallel subcarrier modulation signals are then subjected to an inverse discrete Fourier transform (IFFT) to obtain $N$ time-domain sampling signals, represented as $s_0 \sim s_{N-1}$. Here, the signal of the $n$ OFDM modulation at the $k$ IFFT output sampling point in the time domain is represented as $s_{n,k}$, which can be expressed as:

$$s_{n,k} = \frac{1}{\sqrt{N}}\sum\nolimits_{k=0}^{N-1} S_{N,K}e^{\frac{j2\pi nk}{N}} \tag{9}$$

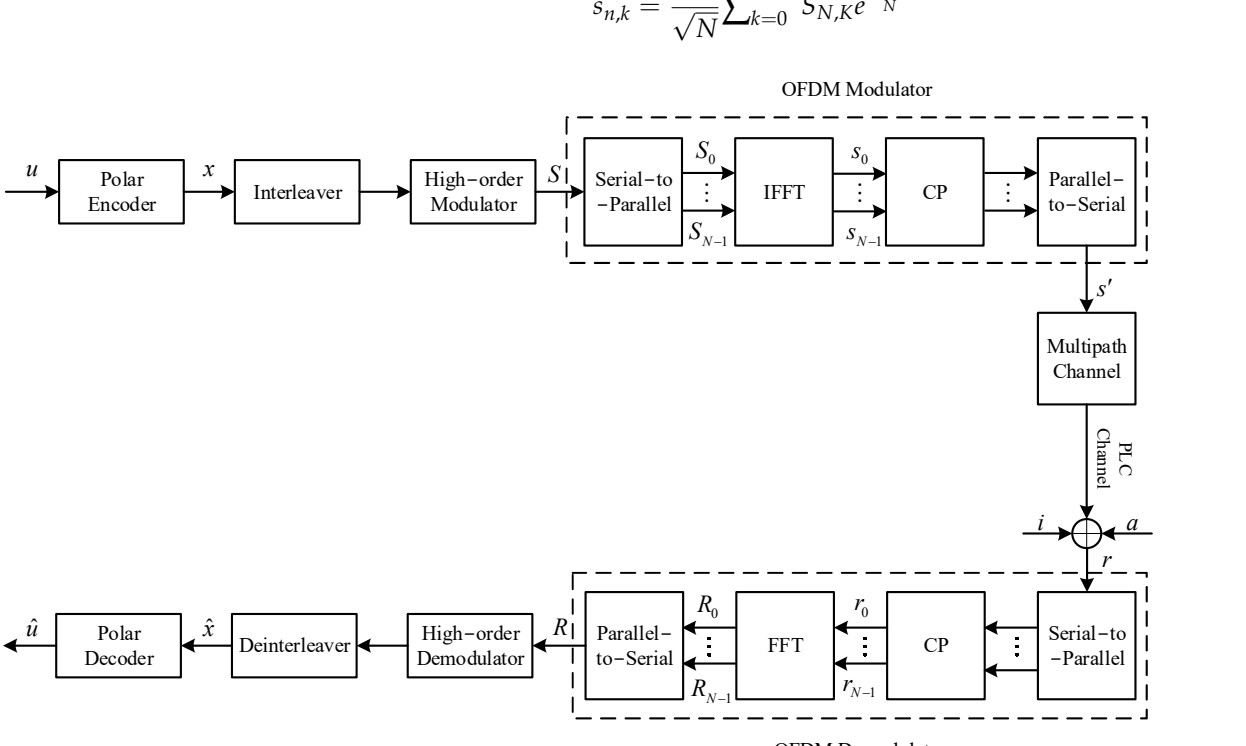

**Figure 2.** PLC polarity encoding system based on OFDM.

After windowing, adding a cyclic prefix (CP), and digital-to-analog conversion, the continuous OFDM modulation signal $s'$ is generated and transmitted to the PLC channel. The continuous OFDM modulation signal is generated and transmitted to the PLC channel, represented as $s' = F_M\widetilde{S}$. Here, $\widetilde{S}$ is a $M \times 1$ zero-padding signal vector, and $F_M$ is the IFFT matrix, represented as $\frac{1}{\sqrt{M}}e^{\frac{j2\pi mk}{M}}$, $\{m = 0, 1, \ldots, M - 1\}$. At the receiver, the received signal $r$ is usually disrupted by additive Gaussian white noise $a$ and impulse noise $i$, which is represented as $r = HF_M\widetilde{S} + a + i$, where $H$ is the cyclic matrix of the multipath channel. Then, a similar OFDM demodulation is performed according to the transmitting end to realize the fast Fourier transform (FFT), and the symbol sequence is obtained:

$$R_k = \frac{1}{\sqrt{N}}\sum\nolimits_{n=0}^{N-1} r_n e^{\frac{j2\pi nk}{N}} \tag{10}$$

Finally, the output data is obtained after the inverse operations corresponding to the structure of the transmitter.

### 3.2. Multipath Channel Model

Due to the impedance mismatch between the inherent impedance of low-voltage power lines and the impedance of a large number of connected electrical devices, signal multipath propagation occurs at certain nodes. In this paper, we adopt the multipath channel transmission model proposed in [23] to establish the PLC channel model, and the frequency response is given by:

$$H(f) = \sum_{i=1}^{K} g_i \cdot A(f, d_i) \cdot e^{-j2\pi f \tau_i} \tag{11}$$

where $K$ is the number of paths; $g_i$ is the weighting coefficient representing the product of the reflection and transmission factors along the $i$ path, $|g_i| \leq 1$, $d_i$ is the propagation distance of the $i$ path, $\tau_i$ is the multipath delay of the $i$ path, represented as $\tau_i = \frac{d_i \cdot \sqrt{\zeta_r}}{c_0}$, where $\zeta_r$ is the dielectric constant and $c_0$ is the speed of light. The attenuation factor of the $i$ path that varies with frequency $f$ and path propagation distance is calculated by:

$$A(f, d_i) = e^{-\alpha(f) \cdot d_i} = e^{-(\alpha_0 + \alpha_1 \cdot f^k) d_i} \tag{12}$$

where $\alpha_0$ and $\alpha_1$ are two attenuation constants, and $k$ is the exponent of the attenuation factor.

### 3.3. PLC Channel Noise Model

The noise situation in actual PLC channels is extremely complex and is mainly divided into two parts: background noise and impulse noise. The background noise includes colored noise, narrowband noise, and periodic impulse noise that is asynchronous with the power frequency. It is usually more stable, has low power, and its intensity does not change dramatically over time. The impulse noise includes periodic impulse noise synchronized with the power frequency and burst impulse noise. It is highly random, with high impulse intensity that decays rapidly within a short period of time, and its power spectral density increases as the frequency decreases. The superposition of this noise results in severe interference to the PLC channel and poor reliability of data transmission. The Middleton class-A model, as a non-Gaussian noise model, is often used to characterize the noise characteristics of the PLC channel, but its simulated impulses are independent sampling points, which do not conform to the physical characteristics of impulse noise and cannot reproduce the randomness of impulse occurrence in real scenes. Therefore, this paper uses a multi-damping sinusoidal function superimposed to simulate impulse noise and the modeling of the single impulse noise waveform function is as follows:

$$n_k(t) = E_k \cdot e^{\frac{-t}{\lambda}} \cdot sin[2\pi f(t) + \varphi] \cdot u(t) \tag{13}$$

where $E_k$ is the impulse amplitude; $\lambda$ is the impulse time constant, which is about one-fifth of the impulse width $t_{k,w}$; $u(t)$ is the unit step function; $\varphi$ is the initial phase, $\varphi \in (0, 2\pi)$; $f$ is the pseudo frequency of the decaying sine function described by the Weibull distribution at frequencies between $0 \sim 30$ MHz. The probability density of the Weibull distribution is as follows:

$$f(x) = abx^{b-1}e^{-ax^b} \tag{14}$$

where $a$ and $b$ are constants. When $f < 3$ MHz, the optimal values of $a$ and $b$ are $5 \times 10^{-7}$ and 6.07, respectively. The optimal values of $a$ and $b$ are 0.59 and 2.27, respectively, when $f > 3$ MHz [24].

To characterize the impulse amplitude and width, parameters $\mu$ and $p$ are introduced. $\mu$ is the power ratio between impulse noise and background noise, $\mu = \frac{\sigma_I^2}{\sigma_B^2}$, which gives the impulse amplitude as $E_k = \sqrt{\frac{\sigma_B^2}{\mu}}$, the total noise power is $\sigma^2 = \sigma_I^2 + \sigma_B^2$. $p$ is the probability of impulse noise width occurring in a signal period $t$. If $b(n)$ satisfies the

Bernoulli distribution process, and $p\{b(k) = 1\} = p$, $(k = 0, 1, \dots)$, then the impulse width is $t_{k,w} = \sum_{l=0}^{N-1} b(i)$.

Therefore, the $k$ impulse noises occurring in a signal period $t$ can be represented as:

$$N(t) = \sum_k n_k(t - t_k) \tag{15}$$

where $t_k$ is the time at which the $k$ impulses arrive, which needs to be determined by measuring the data, and can be represented by a random variable to reflect the randomness of impulse noise. The impulse parameter $m$ is introduced to characterize the probability of impulse arrival time $t_k$ within a signal period $t$, $m \in [10^{-3}, 10^{-1}]$. Then, the number $k$ of impulse noise occurrences in one period can be represented as $k = m * t$. Figure 3 shows the simulated PLC random impulse noise waveform generated when $m = 0.02$, and the inset shows the waveform of a single impulse noise generated by (13).

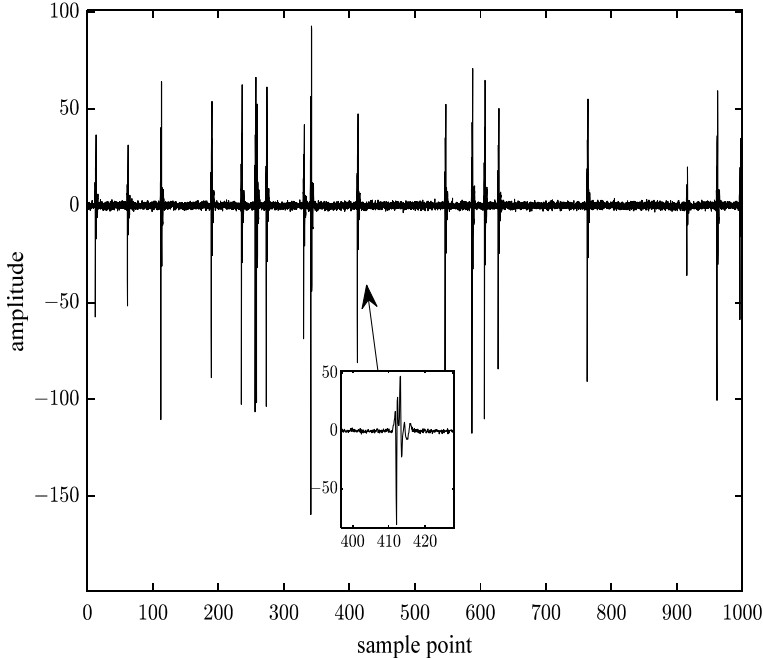

**Figure 3.** Random impulse waveform of the improved noise model with m = 0.02.

## 4. High-Order QAM and APSK Modulation and Demodulation

To improve communication speed and save spectrum resources, high-order modulation methods that combine amplitude and phase, such as Quadrature Amplitude Modulation (QAM) and Amplitude Phase Shift Keying (APSK), have advantages. By combining high-order QAM and APSK modulation with reliable polar code encoding through soft demodulation [25], the performance of PLC systems can be better improved.

### 4.1. QAM Modulation

QAM is a modulation method that combines amplitude and phase modulation, which has better bandwidth utilization compared to single modulation. At time $t$, the $i$-th baseband signal $S_{k,t}(t)$ in the $k$-th QAM modulation symbol can be represented as:

$$S_{k,t}(t) = A_i f(t - iT) \sin(\omega t + \varphi_i) \tag{16}$$

where $i = 1, 2, \dots, N'$, $N'$ is the number of bits included in each QAM modulation symbol; $k = 1, 2, \dots, M'$, $M'$ is the number of constellations in the QAM modulation constellation; $\omega$ is the angular frequency; $\varphi_i$ is the initial phase of the modulation carrier corresponding to the $i$-th baseband signal; $A_i$ is the impulse amplitude of the $i$-th baseband signal; $T$ is the

impulse width of the baseband signal, and $f(t - iT)$ is the rectangular wave generated by the *i*-th baseband signal with an impulse width of $T$.

The RF signal $S_k(t)$ corresponding to the *k*-th QAM modulation symbol is defined as follows:

$$S_k(t) = \sum_{i=1}^{N'} A_i f(t - iT)\sin(\omega t + \varphi_i)$$
$$= I_k(t)sin\omega t + Q_k(t)cos\omega t \tag{17}$$

where $I_k(t) = \sum_{i=1}^{N'} A_i f(t - iT)cos\varphi_i$, $Q_k(t) = \sum_{i=1}^{N'} A_i f(t - iT)sin\varphi_i$, $I_k(t)$ and $Q_k(t)$ are mutually orthogonal at time *t*, and regardless of the time *t* (17) can be expressed in the complex form:

$$S_k(t) = I_k + jQ_k \tag{18}$$

where $I_k$ and $Q_k$, respectively, represent the real and imaginary components of the *k*-th modulation symbol $S_k$.

When $N' = 4$, it is 16QAM modulation, which means that each QAM modulation symbol contains 4 information bits. Figure 4 shows the 16QAM modulation constellation mapping, where each modulation symbol corresponds to a Gray code in the constellation mapping. The 4-bit Gray code sequence $\{b_1, b_2, b_3, b_4\}$ is divided into real and imaginary parts, where $b_1, b_2$ represent the real part of the 16QAM symbol and $b_3, b_4$ represent the imaginary part. The number of real and imaginary bits is the same, which facilitates the subsequent decision calculation for soft demodulation.

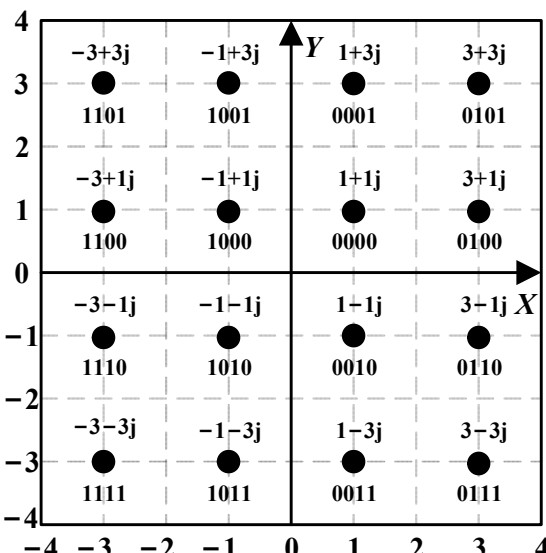

**Figure 4.** 16QAM constellation mapping.

### 4.2. QAM Soft Demodulation

Due to noise interference during channel transmission, the modulation constellation will deviate from its original coordinates, and an effective decision must be made to correctly demodulate the signal. Based on (18), at time *t* corresponding to a certain QAM modulation symbol $S(t) = I(t) + jQ(t)$, the received signal $r(t)$ is:

$$r(t) = h(t)S(t) + g(t) \tag{19}$$

where $h(t)$ is the channel gain, and $g(t)$ is the impulse noise with total noise power $\sigma^2$.

At time *t*, the received signal $r(t)$ contains the QAM modulation symbol $S(t)$, and the error probability of a certain constellation symbol *s* is given by:

$$P[r(t)|S(t) = s] = \frac{1}{\sqrt{2\pi\sigma^2}}\exp\left(-\frac{|r(t) - h(t) \cdot s|^2}{2\sigma^2}\right) \tag{20}$$

By selecting $g(t) = h(t)g_0(t)$, then $r(t) = h(t)[S(t) + g_0(t)] = h(t)y(t)$, and (20) can be simplified as follows:

$$P[r(t)|S(t) = s] = \frac{1}{\sqrt{2\pi\sigma^2}}\exp\left(-\frac{h^2(t)|y(t) - s|^2}{2\sigma^2}\right) \tag{21}$$

To obtain LLR for each bit of the received symbol $r(t)$, the LLR of the $i$-th bit of a symbol in a QAM mapping can be defined as:

$$LLR(b_i) = \log\left(\frac{P(r(t)|b_i = 0)}{P(r(t)|b_i = 1)}\right) \tag{22}$$

By simplifying using Bayes' formula, for $M'$ QAM modulation symbols, the arrangement of 0 bits in the $i$-th symbol is in the first half of the $M'$ modulation symbols and the arrangement of 1 bit is in the second half, expressed as:

$$LLR(b_i) = \log\left(\frac{\sum_{j=1}^{\frac{M'}{2}} {}_{s \in S_{i,j}^{(0)}} P(r(t) \mid y(t) - s)}{\sum_{j=\frac{M'}{2}+1}^{M'} {}_{s \in S_{i,j}^{(1)}} P(r(t) \mid y(t) - s)}\right) \tag{23}$$

The symbols $S_{i,j}$ are linearly processed according to the $I, Q$ branches, where $S_{I,i}^{(0)}, S_{I,i}^{(1)}$ are the modulation symbols for the $i$-th bit in the $I$ branch when it is 0 and 1, respectively, and $S_{Q,i}^{(0)}, S_{Q,i}^{(1)}$ are the modulation symbols for the $i$-th bit in the $Q$ branch when it is 0 and 1, respectively. Then, the LLR of the $i$-th bit $b_{I,i}$ in the $I$ branch of the received symbol $r(t)$ is given by:

$$LLR(b_{I,i}) = \log\left(\frac{\sum_{j=1}^{\frac{\sqrt{M'}}{2}} {}_{s \in S_{I,i,j}^{(0)}} P(r(t) \mid y(t) - s)}{\sum_{j=1}^{\frac{\sqrt{M'}}{2}} {}_{s \in S_{I,i,j}^{(1)}} P(r(t) \mid y(t) - s)}\right) \tag{24}$$

By substituting (21) into (24), we obtain:

$$LLR(b_{I,i}) = \log\left(\frac{\sum_{j=1}^{\frac{\sqrt{M'}}{2}} \exp\left(-\frac{h^2\left|y_i - S_{I,i,j}^{(0)}\right|^2}{2\sigma^2}\right)}{\sum_{j=1}^{\frac{\sqrt{M'}}{2}} \exp\left(-\frac{h^2\left|y_i - S_{I,i,j}^{(1)}\right|^2}{2\sigma^2}\right)}\right) \tag{25}$$

Similarly, the LLR of the $i$-th bit $b_{Q,i}$ in the $Q$ branch of the received symbol $r(t)$ can be obtained as:

$$LLR(b_{Q,i}) = \log\left(\frac{\sum_{j=1}^{\frac{\sqrt{M'}}{2}} \exp\left(-\frac{h^2\left|y_i - S_{Q,i,j}^{(0)}\right|^2}{2\sigma^2}\right)}{\sum_{j=1}^{\frac{\sqrt{M'}}{2}} \exp\left(-\frac{h^2\left|y_i - S_{Q,i,j}^{(1)}\right|^2}{2\sigma^2}\right)}\right) \tag{26}$$

Finally, the correct bit decision values for the $i$-th bit in the $I, Q$ branches of $r(t)$ can be obtained by applying (27) to (25) and (26), respectively:

$$\begin{cases} b_{I,i} = 0, \; if \; LLR(b_{I,i}) \geq 0 \\ b_{I,i} = 1, \; if \; LLR(b_{I,i}) < 0 \\ b_{Q,i} = 0, \; if \; LLR(b_{Q,i}) \geq 0 \\ b_{Q,i} = 1, \; if \; LLR(b_{Q,i}) < 0 \end{cases} \tag{27}$$

Therefore, for the real part bit sequence of the 16QAM constellation mapping symbols in Figure 4, the symbol positions corresponding to the real part bit $b_1 = 0$ are 1 and 3, and the symbol positions corresponding to the real part bit $b_1 = 1$ are $-1$ and $-3$; the symbol positions corresponding to the real part bit $b_2 = 0$ are $-1$ and 1, and the symbol positions corresponding to the real part bit $b_2 = 1$ are $-3$ and 3. According to (25), the LLR of the first bit $b_1$ in the received symbol $y$ is:

$$LLR(b_1) = \log\left(\frac{\exp\left[-\frac{h^2(y-1)^2}{2\sigma^2}\right] + \exp\left[-\frac{h^2(y-3)^2}{2\sigma^2}\right]}{\exp\left[-\frac{h^2(y+3)^2}{2\sigma^2}\right] + \exp\left[-\frac{h^2(y+1)^2}{2\sigma^2}\right]}\right) \tag{28}$$

Similarly, the LLR of the second bit $b_2$ in the received symbol $y$ is:

$$LLR(b_2) = \log\left(\frac{\exp\left[-\frac{h^2(y-1)^2}{2\sigma^2}\right] + \exp\left[-\frac{h^2(y+1)^2}{2\sigma^2}\right]}{\exp\left[-\frac{h^2(y-3)^2}{2\sigma^2}\right] + \exp\left[-\frac{h^2(y+3)^2}{2\sigma^2}\right]}\right) \tag{29}$$

Since the mapping numbers for the real and imaginary bits are the same, the LLRs of the imaginary bits can be obtained directly from the results of the real bits. That is, the LLRs of the third bit $b_3$ and the fourth bit $b_4$ in the received symbol $y$ are:

$$\begin{cases} LLR(b_3) = LLR(b_1) \\ LLR(b_4) = LLR(b_2) \end{cases} \tag{30}$$

Then, according to the decision criterion in (27), the $LLR(b_1)$, $LLR(b_2)$, $LLR(b_3)$, $LLR(b_4)$ can be used to make decisions and demodulate the 4-bit values in the 16QAM modulation symbol.

*4.3. APSK Modulation*

APSK modulation constellation diagram consists of $k$ concentric circles with different radii, with multiple PSK signal points uniformly distributed on each circle. The number of constellation points on the outer circle is greater than that on the inner circle. The signal set $S_k$ formed by the constellation points on the $k$-th concentric circle can be expressed as:

$$S_k = R_k \exp\left[j\left(\frac{2\pi}{m_k}i_k + \theta_k\right)\right] \tag{31}$$

where $R_k$ is the radius of the $k$-th concentric circle, $m_k$ is the number of constellation points on the $k$-th circle, $i_k(i_k = 0, 1, \ldots, m_{k-1})$ is one of the constellation points on the $k$-th circle, and $\theta_k$ is the initial phase of the constellation points on the $k$-th circle.

The constellation diagram of 16APSK consists of two concentric circles, with multiple signal points evenly distributed on each circle. The signal set formed by these points can be represented as:

$$\begin{cases} S_1 = R_1 exp\left[j \cdot \left(\frac{\pi}{2}i_k + \frac{\pi}{4}\right)\right], (i_k = 0, 1, 2, 3) \\ S_2 = R_2 exp\left[j \cdot \left(\frac{\pi}{6}i_k + \frac{\pi}{12}\right)\right], (i_k = 0, 1, \ldots, 11) \end{cases} \tag{32}$$

where $S_1$ represents the signal set formed by the four points on the inner circle, $S_2$ represents the signal set formed by the twelve points on the outer circle, $\frac{\pi}{4}$ and $\frac{\pi}{12}$ are the initial phases of the inner and outer circle signals respectively, and the inner and outer circle radii are $R_1$ and $R_2$ respectively. The ratio $\gamma = \frac{R_2}{R_1}$ is determined by the code rate of the encoding, and the smaller the code rate, the smaller the corresponding value of $\gamma$. As shown in Figure 5, each constellation point in 16APSK can carry 4 bits of information, and Gray coding is used

for constellation mapping, so that adjacent signal points have only a 1-bit difference in their encoding.

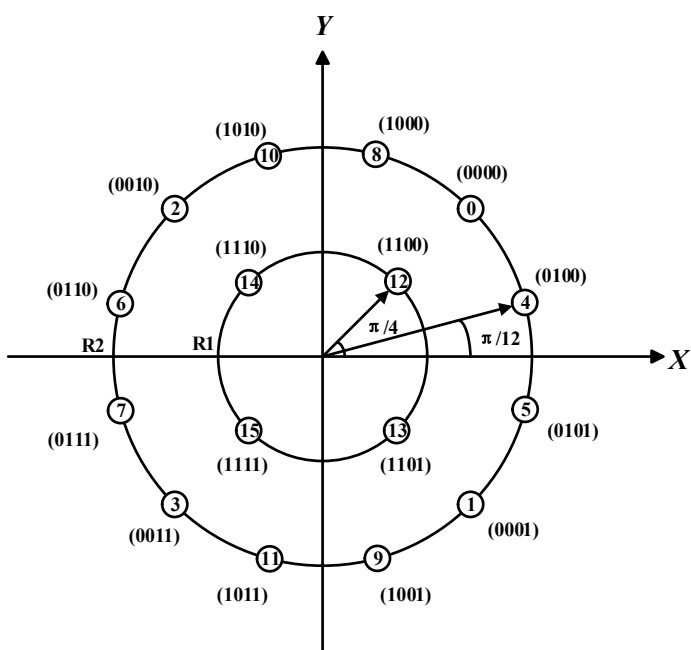

**Figure 5.** 16APSK Constellation Mapping Diagram.

### 4.4. APSK Soft Demodulation

Due to the different constellation structures of APSK and QAM, the real and imaginary parts of APSK are in a nonlinear relationship. Unlike QAM, the modulation symbols of APSK cannot be divided into $I$, $Q$ branch components. When making LLR decisions, the number of bits in the APSK modulation symbols cannot be divided into real and imaginary bits. Instead, the *LLR* of a certain bit position (0 or 1) of all modulation symbols in the constellation diagram must be accumulated. Let $b_{k,i}$ be the $i$-th bit of the $k$-th APSK mapping symbol, and its LLR definition can be expressed as:

$$LLR(b_{k,i}) = \log\left(\frac{\sum_{s=S_i^{(0)}} \exp\left(-\frac{h^2|y_k-s|^2}{2\sigma^2}\right)}{\sum_{s=S_i^{(1)}} \exp\left(-\frac{h^2|y_k-s|^2}{2\sigma^2}\right)}\right) \tag{33}$$

where $k = 1, 2, \ldots, M'$, and $M'$ is the number of modulation symbols in APSK; $i = 1, 2, \ldots, N'$, and $N'$ is the number of bits contained in each APSK modulation symbol; $S_{i,j}^{(0)}$ and $S_{i,j}^{(1)}$ are the sets of all modulation symbols on the constellation diagram in which the $i$-th bit is 0 or 1, respectively; $y_k$ is the $k$-th received symbol; $\sigma^2$ represents the variance of the impulse noise, and $h$ represents the channel gain.

(33) can be simplified by replacing the numerator and denominator sums with their maximum sum as follows [26]:

$$LLR(b_{k,i}) = \frac{1}{2\sigma^2}\left(\min_{s\in S_i^{(1)}}\left(h^2|y_k - s|^2\right) - \min_{s\in S_i^{(0)}}\left(h^2|y_k - s|^2\right)\right) \tag{34}$$

Finally, the LLRs of each bit can be used to make decisions and demodulate the correct bit decisions for the $N'$ bits of the received modulation symbol $y$ by:

$$\begin{cases} b_{k,i} = 0, \ if \ LLR(b_{k,i}) \geq 0 \\ b_{k,i} = 1, \ if \ LLR(b_{k,i}) < 0 \end{cases} \tag{35}$$

Therefore, for the 16APSK modulation symbol containing 4 bits, the LLRs of the 4 bits in the received symbol *y* after being affected by noise can be expressed as:

$$LLR(b_{k,i})_{(i=1,2,3,4)} = \frac{1}{2\sigma^2}\left(\min_{s\in S_i^{(1)}}\left(h^2|y_k-s|^2\right) - \min_{s\in S_i^{(0)}}\left(h^2|y_k-s|^2\right)\right) \quad (36)$$

According to the decision criterion in (35), the LLRs of the 4 bits can be used to make decisions and demodulate the 4-bit values in the 16APSK modulation symbol.

## 5. Performance Simulation and Results Analysis

In order to evaluate the proposed OFDM-PLC system based on polar code encoding, this section conducts simulations and presents simulation results. In the simulation, different noise models, and different high-order modulation methods are compared, and the transmission performance of polar codes in OFDM-PLC systems is studied in different code rates and noise scenarios.

### 5.1. System Parameter Settings

In our simulations, the simulation uses the G3-PLC parameters of the NB PLC standard based on ITU-T G9903, and the physical layer of the G3-PLC protocol uses OFDM modulation to provide robustness against potentially harsh channel conditions. the parameters of polar codes and the OFDM system and their settings in the simulations are shown in Table 1.

**Table 1.** System Parameter Settings.

|  | Parameter | Setting |
|---|---|---|
| | constellation mapping method | BPSK/16APSK/16QAM/16PSK |
| | cyclic prefix length (μs) | 75 |
| | length of cyclic prefix (μs) | 30 |
| | subcarrier spacing *f* | 1.5652 kHz |
| OFDM | sampling rate | 400 kHz |
| | number of subcarriers *N* | 36 |
| | FFT/IFFT size | 256 |
| | OFDM symbol spacing (μs) | 715 |
| | Code length | 512 |
| | Coding rate | $\frac{1}{2}, \frac{5}{8}, \frac{3}{4}$ |
| Polar code | Decoding algorithm | CA-SCL |
| | Decoding list length | 8 |
| | CRC length | 16 |

### 5.2. Giving the BER for Different Intensities of Impulse Noise

The BER performance of polar codes under different impulse models and impulse interference conditions is studied in the above OFDM system using 16APSK modulation. Firstly, the most commonly used Middleton-A class noise model is used, with a fixed Gaussian pulse power ratio of 0.1, which means that the power of the impulse noise is 10 times that of the background noise, and the impulse index *A* in the model, which measures the average number of impulses within one signal cycle, is changed. As shown in Figure 6, with a polar code length of 512, and a code rate of 1/2, when *A* is defined as 0.01, 0.1, and 0.5, it can be seen that as *A* increases, the impulse interference increases, the attenuation slope decreases, and the BER performance decreases. When the BER reaches $10^{-3}$, the Signal-to-Noise Ratio (SNR) of $A = 0.01$ is 2dB higher than that of $A = 0.1$. In order to better fit the background noise characteristics of the PLC channel, an improved noise model is given to characterize more realistic random impulses. The *m* parameters in the improved noise model are set to six values as shown in Figure 7, respectively, and the BER curves using AWGN as the background noise under the same system model are also

given as references. The figure shows that under 16APSK modulation, as $m$ increases, the number of impulse noises increases. When SNR > 10 dB, high-amplitude random impulse noises will reduce the error correction ability of the polar code. As the SNR continues to increase, the BER under the improved noise model tends to flatten, i.e., the error floor phenomenon (error caused by impulse noise only) appears as $m$ increases and increases with $m$.

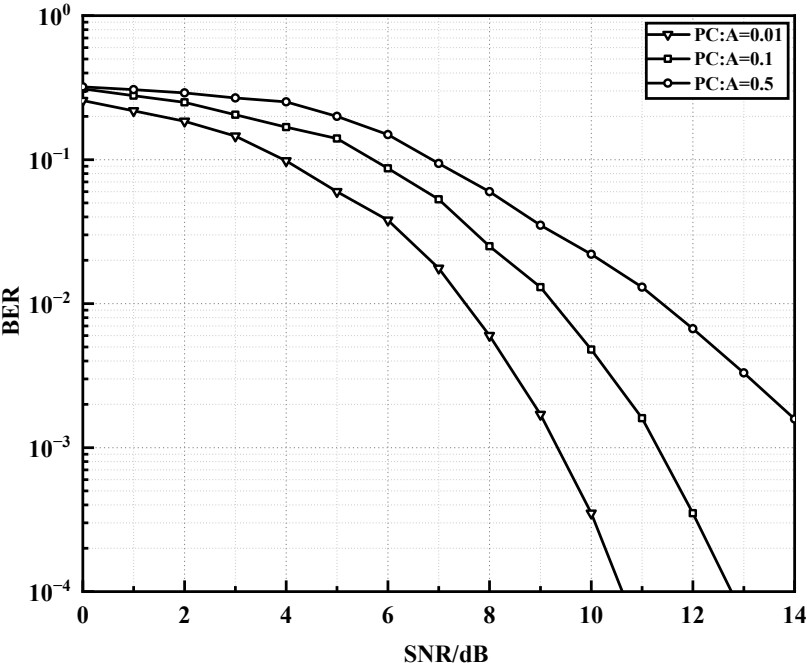

**Figure 6.** BER performance of polar codes under different impulse indices A in Middleton-A class model.

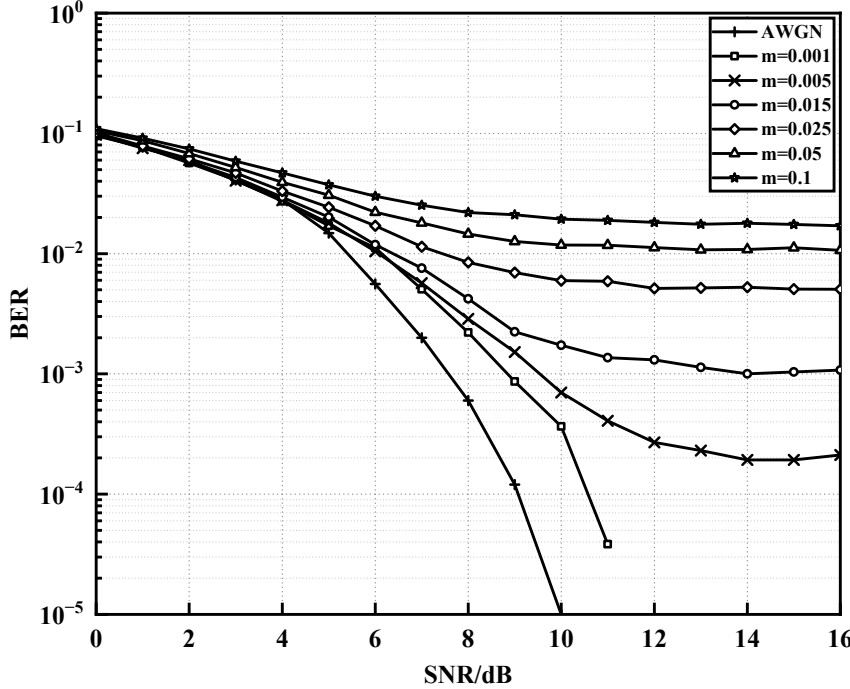

**Figure 7.** BER performance of polar codes under different impulse parameters m in the improved noise model.

*5.3. BER for Different Modulation Methods*

Modulation schemes of the same order are compared under an OFDM-PLC system using the same polar coding method and setting the improved impulse noise parameter to $m = 10 \times 5^{-3}$. Three types of modulation, 16QAM, 16APSK, and 16PSK, were evaluated, and the BER performance is shown in Figure 8. It can be seen that, as the SNR increases, 16QAM has the fastest decrease in BER, followed by 16APSK, and 16PSK has the slowest decrease. 16QAM has slightly better noise resistance than 16APSK and 16PSK in the same order. The performance of 16QAM is superior to 16APSK by about 1dB at BER of $10^{-2}$ and by about 3 dB at BER of $10^{-3}$. As the SNR increases, the performance advantage of 16QAM is more obvious than that of 16APSK under the same polar code coding conditions. Because of the different constellation architectures, the minimum spacing of all adjacent constellation points in the QAM modulated constellation is the same, while the minimum spacing of adjacent constellation points in APSK modulated constellation is only equally spaced on the same circumference, so the distribution of QAM modulated symbols of the same order is more uniform than that of APSK. At the same time, as the modulation order increases, the number of APSK modulation radii of the same order increases, and the phase modulation level difference becomes larger, resulting in more difficult modulation and demodulation and lower modulation performance. However, when the load on the power line changes, channel distortion may occur. The 16QAM is the most sensitive to amplitude distortion because it has the most amplitude values, while 16PSK is the most sensitive to phase distortion because all its signal points are on the same circle, and the phase spacing between adjacent points is minimal. The sensitivity of 16APSK to both types of distortion is between 16QAM and 16PSK, and it is more balanced in terms of noise resistance and distortion resistance, making it more suitable for PLC communication systems.

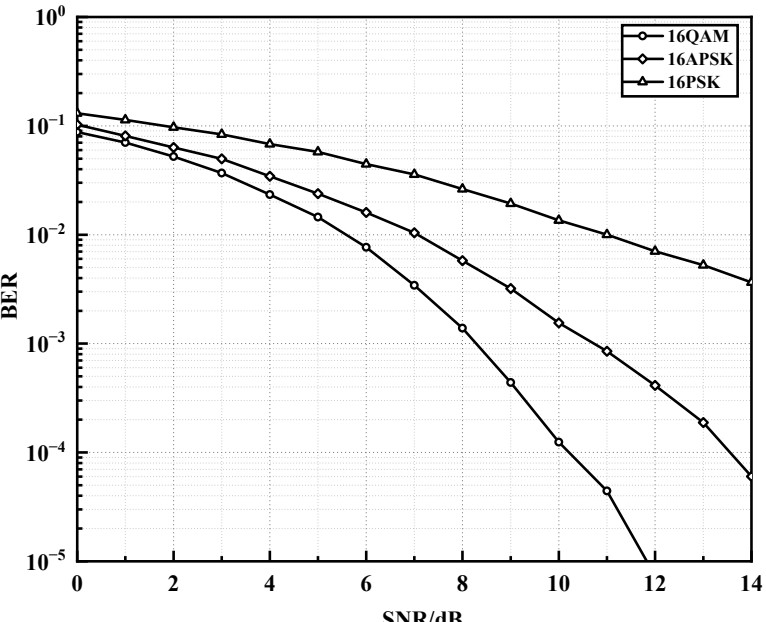

**Figure 8.** BER performance under different modulation schemes.

*5.4. Noise Immunity Using Polar Codes and LDPC Codes*

Finally, by simulating the analysis in Figures 6–8, the transmission performance of different coding schemes under the OFDM-PLC-based system is focused on by selecting the high-order 16QAM modulation in the final system model and setting the improved impulse noise parameter $m = 10 \times 5^{-3}$. The BER performance of the polar and LDPC code schemes with three different coding rates (1/2, 5/8, and 3/4), respectively, and code lengths of 512 are compared under Gaussian white noise and improved impulse noise, where the LDPC code scheme is a combination of Approximate Lower Triangular (ALT) encoding and

BP algorithm decoding has a maximum number of iterations of 50. As shown in Figure 9a, with the same code length, the decoding performance of both polar codes and LDPC codes deteriorates as the code rate increases under Gaussian white noise only. At BER of $10^{-3}$, the LDPC code and polar code with a code rate of 1/2 both have approximately 1 dB better performance than those with a code rate of 3/4. In the high SNR region, such as at a code rate of 3/4 and SNR > 3 dB, the decay slope of the polar code scheme increases significantly, and its error correction performance is enhanced. Similarly, for code rates of 1/2 and 5/8, the advantages of the polar code scheme over the LDPC scheme become more pronounced with increasing SNR. As shown in Figure 9b, when improved impulse noise is added on top of Gaussian white noise, at BER of $10^{-3}$ and code rate of 1/2, both polar code and LDPC code schemes show approximately 2 dB performance reduction compared to that under Gaussian white noise only. However, at BER of $10^{-2}$ and code rate of 5/8, the performance of LDPC codes with added impulse noise is reduced by more than 2 dB over Gaussian white noise only, which is greater than that of PC codes under the same conditions, and the trend becomes more pronounced as the code rate and SNR increase. The decay slope of the polar code scheme under impulse noise is greater than that of the LDPC code, and its error correction capability is significantly enhanced compared to that of the LDPC code scheme. Therefore, the polar code scheme shows better suppression of impulse noise than the LDPC code scheme.

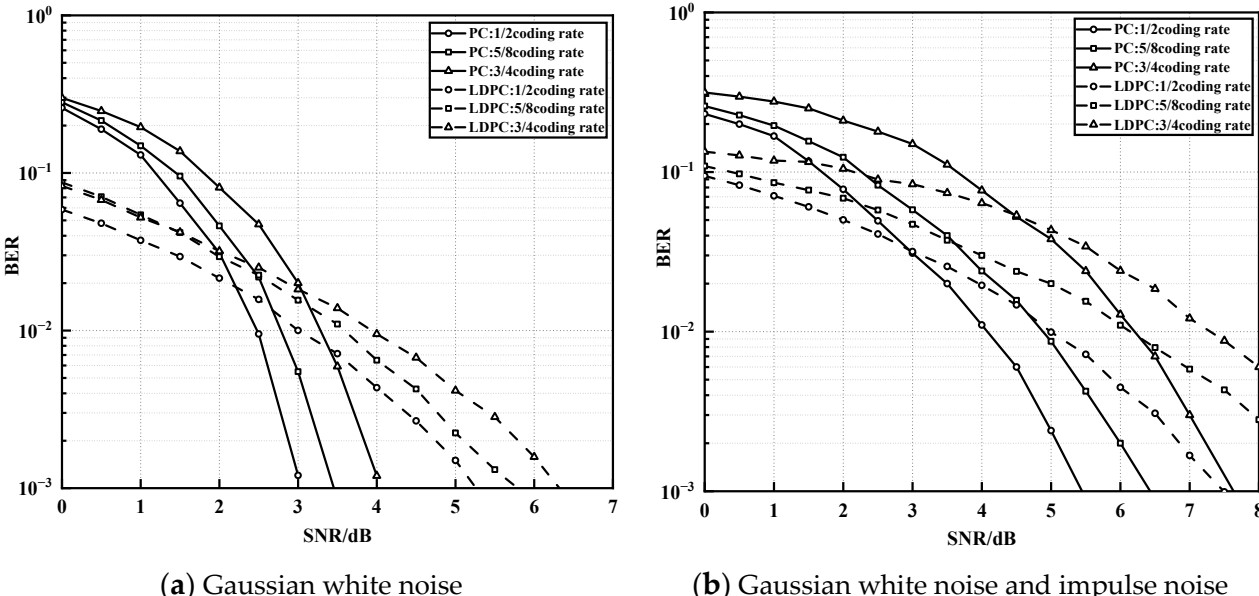

(**a**) Gaussian white noise

(**b**) Gaussian white noise and impulse noise

**Figure 9.** BER Performance of PC and LDPC at Different Code Rates.

## 6. Conclusions

This paper proposes an OFDM-PLC impulse noise control system based on polar code encoding, using a multi-damping sinusoidal function superimposed to improve the modeling of impulse noise. By combining coding and modulation techniques, a polar code module is introduced into the OFDM-PLC system and the BER performance of the polar code-based system under high-order QAM and APSK modulation is investigated. Simulation results demonstrate that the proposed polar coding scheme has lower complexity compared to the LDPC coding scheme, and can reduce the error rate during PLC transmission of impulse noise interference. This is of significant practical importance for actual PLC systems.

**Author Contributions:** Software, Y.W.; Writing—original draft, Z.W.; Writing—review & editing, R.C. All authors have read and agreed to the published version of the manuscript.

**Funding:** This work was financially supported by the Research Innovation Program for College Graduates of Jiangsu Province (No. SJCX23_1204), the Science and Technology Special Fund (Social Development R&D Plan) of Jiangsu Province (SBE2022740885).

**Institutional Review Board Statement:** Not applicable.

**Informed Consent Statement:** Not applicable.

**Data Availability Statement:** Not Applicable, the study does not report any data.

**Conflicts of Interest:** The authors declare no conflict of interest.

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
