# Peer review of "Polar Codes and a M-ary Modulation-Based OFDM-PLC System"

_information, doi:10.3390/info14070360_

Round 1

Reviewer 1 Report

Dear Authors,

There are many basic information: 

1. In section 4, we have details of QAM and APSK modulation and demodulation derivations but these findings are not used in the later stage. Then, I do not see reason why these things are there.

2. There is no connection between chapter 3 and chapter 4. There should be connection. There should be a link between them.

3. OFDM is used in Section 5 simulation and result analysis but never used before during the mathematical analysis. Results are not linked with previous chapters.

4. Figure 6, 7 and 8 are just BER performance analysis in different cases and scenarios and have nothing to do with final results.

5. In Section 5.4, BER Performance of PC and LDPC at different Code Rates in two types of noise is presented but there is no linkage with channel introduced in previous section, modulation, and demodulation.

6. There are many versions of LDPC codes and also different versions of coding and decoding but there is no mentioned of anything in the paper. It is not clear how can you compare the results. 

7. The results are not linked with title of the paper.

Best of luck 

Reviewer 2 Report

This paper analyzes the standard BER v/s SNR performance metric for PLC channels using polar codes for higher order modulation schemes. Since this is a performance analysis paper similar to a review paper, the review is focused on the accuracy and extent of literature review, rather than novelty of the presented work. Following are some comments on the paper. Incorporating these can improve the quality of the paper further.

Firstly, this is a well organized paper. The reviewer appreciates the organization and presentation of the paper.

The channel and noise model used in the paper is different from most typical considerations (fading for channel and Middleton noise for impulsive noise). The channel model considered in this paper is correct and the fading model that is usually applied for PLC is actually incorrect (although that is accurate for wireless channels). The noise model introduced in this paper is different from Middleton Class A model. The multi-damping sinusoidal model is accurate and the presented result looks accurate to typical measured impulse noises. However, is this noise model universally applicable to different scenarios? The authors could provide more references to justify the chosen model and to their claims that the Middleton model is generally non-reproducible. 

The reviewer recognizes the comparison of QAM results with APSK. However, it doesn’t look like APSK is used in typical PLC modems or systems. Can the authors provide examples of modem transmitters and receivers that use APSK? Which standard has specified this? 

In the simulation section, a block diagram of the transceiver needs to be given for a reader to appreciate the work, and also for the reviewer to accurately judge whether the simulation model is correct or not.

The BER vs SNR results in all three figures (6-8) seem a little odd. The error rates for about 0-6 SNR looks to be close to 1-10% after error correction. This is unusually high. The authors must comment on the results and compare them with other existing arts to show that the numbers are reasonable. Otherwise, a 10% bit error rate (which would lead to a much higher frame error rate) is almost unworkable for typical systems.

Overall, this is a good review paper. If the problems that are highlighted above are solved, this paper has the potential to be accepted.

No major changes suggested.

Reviewer 3 Report

The paper lacks essential parameters of the considered codes and decoding algorithms, i.e. code length, CRC length, list size, number of iterations. 

I strongly suggest also to show the decoding complexity of the considered codes in terms of the number of arithmetic operations, and make sure that polar and LDPC codes have comparable decoding complexity

Round 2

Reviewer 1 Report

Dear Authors,

Well done.

Reviewer 2 Report

The authors have addressed all comments by the reviewer in the response letter and made appropriate changes to the manuscript. This has substantially improved the quality of the paper and is ready to be accepted.

None